# Engineering Microorganisms to Produce Bio-Based Monomers: Progress and Challenges

Chenghu Chen [1,2], Xiulai Chen [1,2], Liming Liu [1,2], Jing Wu [3] and Cong Gao [1,2,*]

1    State Key Laboratory of Food Science and Technology, Jiangnan University, Wuxi 214122, China
2    International Joint Laboratory on Food Safety, Jiangnan University, Wuxi 214122, China
3    School of Life Sciences and Health Engineering, Jiangnan University, Wuxi 214122, China
*    Correspondence: conggao@jiangnan.edu.cn

**Abstract:** Bioplastics are polymers made from sustainable bio-based feedstocks. While the potential of producing bio-based monomers in microbes has been investigated for decades, their economic feasibility is still unsatisfactory compared with petroleum-derived methods. To improve the overall synthetic efficiency of microbial cell factories, three main strategies were summarized in this review: firstly, implementing approaches to improve the microbial utilization ability of cheap and abundant substrates; secondly, developing methods at enzymes, pathway, and cellular levels to enhance microbial production performance; thirdly, building technologies to enhance microbial pH, osmotic, and metabolites stress tolerance. Moreover, the challenges of, and some perspectives on, exploiting microorganisms as efficient cell factories for producing bio-based monomers are also discussed.

**Keywords:** bioplastics; monomer; synthetic biology; metabolic engineering

## 1. Introduction

Current plastics are mainly synthesized or semi-synthesized from fossil fuels [1]. During this process, greenhouse gas emissions are associated with every stage of the production, application, recycling, and incineration of plastic [2]. As a result, plastic pollution has become one of the most important environmental problems. To eliminate the dependence on fossil fuels and reduce environmental pollution, bioplastics, a substitute for fossil-based plastics, have attracted increasing attention [3–6]. Bioplastics are bio-based and/or biodegradable polymers made from renewable resources, such as cellulose, starch, lignin, etc. [7–9]. Compared with fossil-based plastics, bio-based plastics have a lower carbon footprint and exhibit favorable material properties, playing a key role in creating a low-carbon circular economy [10]. In addition, bioplastics can be recycled into raw materials, which improves the efficiency of resource recycling and makes a crucial contribution to resource conservation and environmental protection [11].

According to the annual report (2021) from European Bioplastics, the representative bioplastics included polybutylene adipate terephthalate (PBAT), polylactic acid (PLA), polybutylene succinic acid (PBS), Polyamide (PA), polytrimethylene terephthalate (PTT), and polyethylene terephthalate (PET), which account for about 65% of the total bioplastics market share. Among them, the common monomers included 1,4-butanediol (1,4-BDO), 1,3 propanediol (1,3-PDO), adipic acid, succinic acid, cadaverine, glutaric acid, lactic acid, and terephthalic acid, etc. (Table 1).

**Table 1.** Applications and market potential of bioplastics monomers.

| Monomers | Applications | Market | References |
|---|---|---|---|
| 1,4-butanediol | Medical treatment, Food, Chemical industry, Materials | USD 6.19 billion | [12–14] |
| 1,3 propanediol | Chemical industry, Materials | USD 776.3 million | [15–17] |
| Succinic acid | Agriculture, Green solvents, Pharmaceuticals, Biodegradable plastics, Materials | 245,000 tons | [18,19] |
| Adipic acid | Chemical industry, Materials | USD 6 billion | [20–22] |
| Cadaverine | Nylon, Chelating agents, Materials | 220 million | [23–28] |
| Glutaric acid | Fine chemicals, Monomers, Building blocks, Materials | NM | [29–31] |
| Lactic acid | Food, Pharmaceutical, Chemical, Cosmetic industries | 2 million tons | [32–35] |

NM, not mentioned.

Microbial fermentation has the advantages of being relatively mild, environmentally friendly, and easy to operate, which has broad applications [36–39]. The annual market of bio-plastics is expected to increase to 18% by 2025 [40]. With increasing gene manipulation tools being constructed, various microbial cell factories have been developed to produce high-value chemicals [41,42]. Some dicarboxylic acid such as succinic acid has already been commercialized for production using microbial cell factories [43]. However, it is still challenging to realize the industrial-scale bioproduction of many other monomers due to their low economic feasibility [44]. The development of efficient bio-based monomers is hampered by three main obstacles [45]: First, the low utilization efficiency of raw materials. Increasing the utilization efficiency of renewable raw materials during monomer production is critically important for the avoidance of food crops or restructuring of fermentation infrastructure [46]; Second, poor monomer synthetic efficiency. Most of the synthetic pathways for microbial monomer production are complex and involved multi-step reactions [47]; Third, weak environmental tolerance. Various factors such as pH, osmotic, and high concentrations of metabolites can cause stress on the performance of large-scale microbial monomer production [48]. The key performance indices, including production titer, productivity, and yield, together with the host strains and metabolic engineering strategies employed are summarized in Table 2.

**Table 2.** Metabolic engineering strategies on microorganisms for monomer production.

| Monomers | Strains | Tools or Strategies | Features | References |
|---|---|---|---|---|
| Cadaverine | *Methylosinus trichosporium* | Introducing lysine decarboxylase, aspartokinase, and meso-diaminopimelate decarboxylase | 283.63 mg·L$^{-1}$ | [49] |
| | *Corynebacterium glutamicum* | Cell surface display technique using PorH anchor protein, introducing xylose assimilation pathway | 11.6 mM | [50] |
| | *Escherichia coli* | Rational engineering, Protein-directed evolution | 418 g·L$^{-1}$ | [51] |
| | *E. coli* | Combining directed evolution and computation-guided virtual screening | 160.7 g·L$^{-1}$ | [52] |
| | *E. coli* | genomic analysis, constructing 67 genes-repressing sRNAs | 13.7 g·L$^{-1}$ | [53] |
| | *E. coli* | Introducing zwitterionic peptides into lysine decarboxylase to promote correct folding, | Doubled enzymatic activity | [54] |
| | *E. coli* | In situ $CO_2$ recapture technology, modifying bioreactor to recapture the $CO_2$ | 0.99 mol·mol$^{-1}$ lysine | [55] |

**Table 2.** *Cont*.

| Monomers | Strains | Tools or Strategies | Features | References |
|---|---|---|---|---|
| Adipic acid | *Pseudomonas taiwanensis* | Enzyme mining, cascade reaction design, metabolic optimization | $10.2 \text{ g·L}^{-1}$ | [56] |
| | *E. coli* | Chemical solvent treatment and physical crushing methods | $0.39 \text{ g·g}^{-1}$ glucose | [57] |
| | *Saccharomyces cerevisiae* | Three-stage fermentation process optimization, screening heterologous enzymes | Directly produce adipic acid using glucose | [58] |
| | *E. coli* | Establish an oxygen-dependent dynamic regulation system | increase adipic acid titer by 41.62-fold | [59] |
| | *S. cerevisiae* | Overexpressing specific multidrug resistance transporters | Reducing toxic side effects of adipic acid on *S. cerevisiae* cells | [60] |
| Medium-chain α, ω-dicarboxylic acids | *E. coli* | Modularizing the β-oxidation pathway and ω-oxidation pathway, adaptive evolution | $15.26 \text{ g·L}^{-1}$ | [61] |
| Malate | *E. coli* | Integrating synthetic $CO_2$ fixation and $CO_2$ mitigation modules | $1.48 \text{ mol·mol}^{-1}$ glucose | [62] |
| | *E. coli* | Overexpressing the ATP-generating phosphoenolpyruvate carboxykinase combined with the $CO_2$ fixation pathway | $CO_2$ fixation efficiency was increased by 110% | [63] |
| 1,3-PDO | *E. coli* | Protein engineering and the expression of native alcohol dehydrogenase | directly produce 1,3-PDO from the sugar | [64] |
| | *E. coli* | Endogenous glycerol assimilation pathway was eliminated, and mannitol was fed as a co-substrate | $0.76 \text{ mol·mol}^{-1}$ glycerol | [65] |
| | *E. coli* | Cell surface display technique | Direct production of 1,3-PDO from starch | [66] |
| | *Clostridium butyricum* | Optimizing the culture conditions | $70.1 \text{ g·L}^{-1}$ | [67] |
| | *Klebsiella pneumoniae* | Knocking out glucose transporter Crr | $78 \text{ g·L}^{-1}$, glycerol conversion rate reaching 59.5% | [68] |
| | *E. coli* | Fine-tune hybrid pathways, match the energy demand, carbon coordination | $22.66 \text{ g·L}^{-1}$ | [69] |
| | *Vibrio natriegens* | Deleting the global transcriptional regulators, transcriptomics analysis | $0.50 \text{ mol·mol}^{-1}$ glycerol | [70] |
| | *C. butyricum* | Combinational chemical (NTG) and plasma-based mutagenesis (ARTP) process | $37 \text{ g·L}^{-1}$, it was 29.48% higher than that of the wild strain | [15] |
| | *Lactobacillus reuteri* | Electron beam irradiation mutagenesis irrelevant | $93.2 \text{ g·L}^{-1}$, it was 34.6% higher than that of the wild strain | [71] |

**Table 2.** *Cont.*

| Monomers | Strains | Tools or Strategies | Features | References |
|---|---|---|---|---|
| cis, cis-muconic acid | *E. coli* | Parallel metabolic pathway engineering | 4.09 g·L$^{-1}$, 0.31 g·g$^{-1}$ glucose | [72] |
| | *E. coli* | Sensor-regulator and RNAi-based bifunctional dynamic switch | 1.8 g·L$^{-1}$ | [73] |
| Lactic acid | *Trichoderma reesei* and *Rhizopus delemar* | Co-culture strategy | Producing lactate from microcrystalline cellulose | [74] |
| | *Pseudomonas putida* and *Bacillus coagulans* | Developing a synthetic consortium | 35.8 g·L$^{-1}$ | [75] |
| | *E. coli* | Screening xylose catabolic operon | a 50% increase in titer than that of the wild strain | [76] |
| | *Lacobacillus manihotivorans* | Recruited for simultaneous saccharification and fermentation of the substrate | 18.69 g·L$^{-1}$ | [77] |
| | *Lactococcus lactis* | The mutation of key proteins and hosts, screening the hetero-molecular chaperone protein Dnak mutant | Stress tolerance and lactic acid production can be greatly improved | [78] |
| | *L. lactis* | Introduce protective agents, add antioxidants | improve the intracellular pH of *L. lactis*, improving acid stress resistance | [79] |
| | *S. cerevisiae* and *E. coli* | Improve proton conversion in cell metabolism or engineer some acid-tolerance genes | Exhibited good cell growth and productivity under high concentrations of lactic acid | [80,81] |
| 2,5-Furandicarboxylic acid | *Synechococcus elongatus* and *P. putida* | Co-culturing engineered, surface engineering | The FDCA yield is elevated to almost 100% | [82] |
| Terephthalic acid | *E. coli* | Oleyl alcohol was recruited as an organic phase for biphasic microbial transformation | 6.9 g·L$^{-1}$ | [83] |
| 2-pyrone-4,6-dicarboxylic acid | *E. coli* | Chemo–microbial hybrid process, microwave-assisted, whole-cell conversion strategy | A yield of up to 96% | [84] |
| Medium-chain polyhydroxyalkanoate | *P. putida* and *E. coli* | Developing a synthetic consortium | 1.02 g·L$^{-1}$ | [85] |
| Succinic acid | *Actinobacillus succinogenes* | Cell immobilization based on biofilm, biofilm-based cell-immobilized fermentation technology | 18% increased titer compared with that of free cell fermentation | [86] |
| | *E. coli* | Using promoters to co-expression of $CO_2$ transport and fixation genes | 89.4 g·L$^{-1}$ | [87] |
| | *Mannheimia* | Building glycerol and glucose substrate utilization pathways | The reducing equivalents mole generated was doubled | [88] |

<div align="center">**Table 2.** *Cont.*</div>

| Monomers | Strains | Tools or Strategies | Features | References |
|---|---|---|---|---|
| Succinic acid | *E. coli* | ATP-generating phosphoenolpyruvate carboxykinase co-expressed with the xylose utilization pathway | 11.13 g·L$^{-1}$ | [89] |
| | *Mannheimia succiniciproducens* | Malate dehydrogenase from different sources was screened and characterized | 134.25 g·L$^{-1}$ | [88] |
| | *E. coli* | Replacement of high energy consumption pathway | Increase titer by 282% | [90] |
| | *E. coli* | Adaptive laboratory evolution | 27.33 g·L$^{-1}$ | [91] |
| | *E. coli* | Identifying Cus copper efflux system to activate CusCFBA expression to transport Cu(I) out of cells | a 36% increase in biomass | [92] |
| | *E. coli* | Introducing mutations in DNA-dependent RNA polymerase RpoB | 40% increase in cell growth and succinic acid production | [93] |
| | *E. coli* | Global regulation, overexpressing *E. coli* global regulator IrrE | 24.5 g·L$^{-1}$ | [94] |
| | *A. succinogenes* | Add protective agents | Increase the titer by 22% | [95] |
| | *E. coli* | A digital pH-sensing system RiDE was designed for the autonomous control of strain evolution | Rapidly obtaining the desired phenotypes | [96] |
| | *Propionibacterium acidipropionici* | In situ product separation, developing membrane separation coupled fermentation technology | Increase the titer by 48.54% | [97] |
| | *M. succiniciproducens* | Change the fluidity of the cell membrane, membrane engineering | 84.21 g·L$^{-1}$, 1.27 mol·mol$^{-1}$ glucose | [98] |
| | *Yarrowia lipolytica* | overexpressing glycerol kinase GUT1 | Increased cell growth and product synthesis | [99] |
| Glutaric acid | *E. coli* | Introducing the malonic acid utilization pathway | 6.3 g·L$^{-1}$ | [100] |
| | *E. coli* | Stronger promoters and high-copy plasmids, bi-directional cadaverine transporter | 54.5 g·L$^{-1}$ | [101] |
| | *E. coli* | Colloidal chitin cell immobilization strategies | 73.2 g·L$^{-1}$ | [102] |
| | *E. coli* | Plasmid optimization, promoter engineering, and ribosome binding site engineering | 77.62 g·L$^{-1}$ | [103] |
| | *C. glutamicum* | Increasing precursor concentrations by overexpressing key enzymes in a synthetic pathway | 22.7 g·L$^{-1}$ | [40] |
| | *E. coli* | Autonomous circuit targeting byproduct synthetic pathways | Decreased accumulation of acetic acid, lactic acid, and formic acid | [104] |
| | *C. glutamicum* | Genomic analysis, combinational engineering | 105.3 g·L$^{-1}$ | [105] |
| | *E. coli* | Coupling the regeneration and consumption of cofactors or precursors | 54.5 g·L$^{-1}$ | [101] |
| 1,2,4-butanetriol | *E. coli* | Trigger factor, GroEL-GroES, and DnaK-DnaJ-GrpE, introducing trigger factor | 1.01 g·L$^{-1}$ | [106] |

**Table 2.** *Cont.*

| Monomers | Strains | Tools or Strategies | Features | References |
|---|---|---|---|---|
| Malonic acid | *Saccharomyces cerevisiae* | Overexpressing key enzymes in a synthetic pathway | 1.62 g·L$^{-1}$ | [107] |
| glucaric acid | *E. coli* | Synthetic protein scaffold to shorten the spatial distance, constructing a substrate channel | Exhibited a 5-fold improvement | [108] |
| | *E. coli* | Designing protein abundance bifunctional molecular switch | 1.16 g·L$^{-1}$ | [109] |
| Itaconic acid | *Aspergillus niger* | Various dynamic switches design, a low-pH-induced promoter Pgas | 4.92 g·L$^{-1}$ | [110] |
| 2,3-butanediol | *E. coli* | Enforced ATP wasting was introduced to consume ATP | Increase the titer by 10-fold | [111] |
| NM | *E. coli* | Global regulation, random mutagenesis | Increase growth ability by nearly 50% | [112] |

In this review paper, various methods and strategies to improve the efficiency of microbial monomer production are discussed from three aspects: (i) how to realize the efficient utilization of cheap substrates by microorganisms to reduce production costs; (ii) how to improve the efficiency of monomer production in microbial cell factories to improve production performance; and (iii) how to strengthen the environmental tolerance of chassis cells to optimize cell activity. Moreover, the future development direction of bio-based monomers was summarized and highlighted. Metabolic pathways for the production of dicarboxylic acids and diamines are shown in Figure 1.

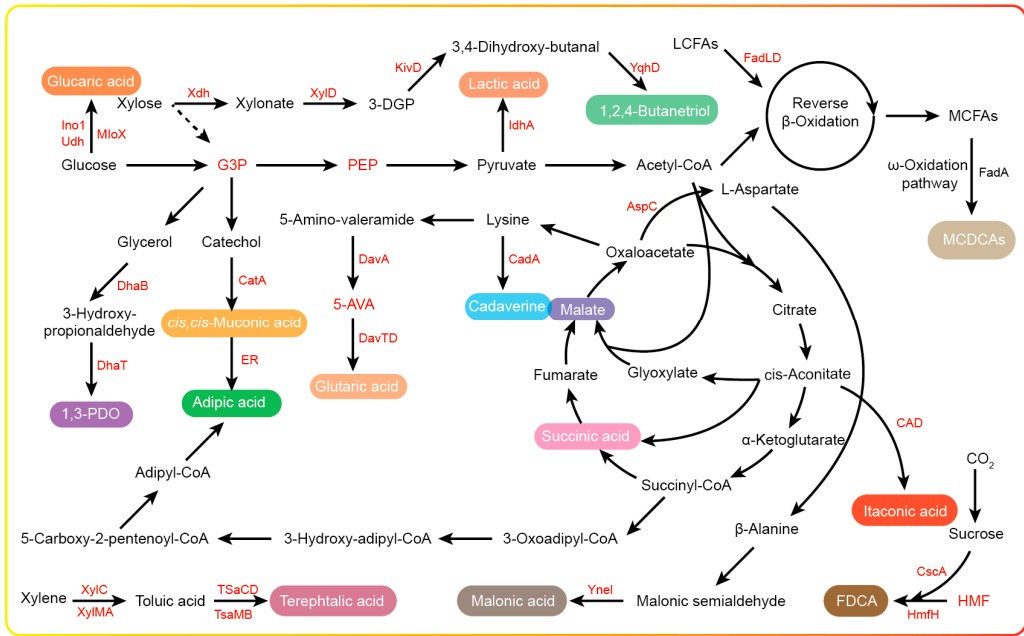

**Figure 1.** Overall metabolic pathways for the production of biobased-monomers from substrates. Biobased monomers including lactic acid, succinic acid, glutaric acid, adipic acid, 1,2,4-butanetriol, MCDCA, Itaconic acid, FDCA, malate, cadaverine, terephthalic, glucaric acid, cis,cis-muconic acid, 1,3-PDO and malonic acid are shown in square boxes. The abbreviations are as follows: CadA, L-lysine decarboxylase I; CatA, catechol 1,2-dioxygenase; DavA, 5-aminovaleramidase; DavD, glutaric semialdehyde dehydrogenase; DavT, 5-aminovalerate aminotransferase; ER, enoate reductase; KivD,

$\alpha$-keto acid decarboxylase; ldhA, lactate dehydrogenase; YneI, succinate semialdehyde dehydrogenase. TsaC, 4-CBALdehydrogenase; TsaD, 4-CBA dehydrogenase; TsaMB, p-toluene sulfonate monooxygenase; Xdh, xylose dehydrogenase; XylC, benzaldehyde dehydrogenase; XylD, xylonate dehydratase; XylMA, xylene monooxygenase; YqhD, alcohol dehydrogenase; DhaB, glycerol dehydratase; DhaT, 1,3-PDO oxidoreductase; AspC, L-aspartate aminotransferase; CAD, cis-aconitate decarboxylase; Ino1, *myo*-inositol-1-phosphate synthase; MIoX, *myo*-inositol oxygenase; Udh, uronate dehydrogenase; 3DGP, 3-deoxy-glycerolpentulsonate; FadD, fatty acyl-CoA synthetase; CscA, sucrose-6-phosphate hydrolase; HmfH, HMF/furfural oxidoreductase; 5-AVA, 5-aminovaleric acid; G3P, D-glyceraldehyde-3-phosphate; PEP, phosphoenolpyruvate; MCFAs, medium-chain fatty acid; LCFAs, long-chain fatty acids; FadL, long-chain fatty acid outer membrane protein; FadA, fatty acid oxidation complex; MCDCAs, medium-chain $\alpha,\omega$-dicarboxylic acids; 1,3-PDO, 1,3 propanediol; FDCA, 2,5-Furandicarboxylic acid; HMF, 5-hydroxymethylfurfural.

## 2. Efficient Utilization of Cheap Substrates

There are abundant renewable resources in nature, such as $CO_2$, lignocellulose, glucose, etc., but the efficient utilization of these substrates is challenging. To solve this issue and improve the synthetic efficiency of engineered strains, three main approaches could be developed, including (i) designing substrate utilization pathways; (ii) enhancing substrate utilization capacity; and (iii) optimizing the substrate transformation process (Figure 2).

### 2.1. Designing Substrate Utilization Pathways

According to the types of metabolic pathways, the design of substrate utilization pathways can be divided into five categories: (i) exploiting endogenous substrate utilization pathway; (ii) building artificial substrate utilization pathway; (iii) combining endogenous and artificial substrate utilization pathways; (iv) building orthogonal substrate utilization pathways; (v) coupling multi-bacterial substrate utilization pathways. All these pathway designs can improve the conversion of cheap substrate into high-value-added bio-based monomers.

### 2.1.1. Exploiting Endogenous Substrate Utilization Pathway

Many bio-based monomers are produced by heterotrophic microorganisms, such as *E. coli* and yeast, but these microorganisms are difficult to use C1 substrates, such as $CO_2$ and methane. By recruiting microorganisms with native utilization capacity of C1 substrates, the utilization pathway can generate a strong metabolic driving force to maximize the carbon flow towards the desired monomer production. For example, the photoautotrophic microbe *S. elongatus* could be engineered to produce lysine by promoter characterization and genome-level metabolic pathway engineering. On this basis, different heterogeneous synthetic pathways were introduced in the engineered strain to achieve the direct photosynthetic production of bio-based monomers (such as cadaverine and glutaric acid) from $CO_2$ [113]. Similarly, an engineered type II methanotroph, *M. trichosporium* OB3b, was engineered for the biosynthesis of cadaverine from methane. By introducing lysine decarboxylase, aspartokinase, and meso-diaminopimelate decarboxylase, the final engineered strain could produce 283.63 mg·$L^{-1}$ of cadaverine directly from methane [49].

### 2.1.2. Building Artificial Substrate Utilization Pathway

For some special substrates, there are no natural substrate utilization pathways for some microorganisms, thus hindering the current progress of developing efficient microbial cell factories to produce desired products. To this end, different artificial pathways can be designed to enable engineered strains to utilize non-metabolic substrates, such as chemical wastes or food wastes, to obtain the desired chemicals. For example, by enzyme mining, multi-enzyme cascade reaction design, and metabolic optimization, a *P. taiwanensis* VLB120 was engineered to utilize cyclohexane to produce 6-hydroxyhexanoic acid and then further decomposed into adipic acid. After pathway enzyme optimization and bioreactor optimization, the final engineered strain could produce 10.2 g·$L^{-1}$ adipic acid with

cyclohexane as substrate [56]. The recycling and reuse of food waste is also a promising alternative to support and promote a circular economy [114]. For instance, by modularizing the β-oxidation pathway and ω-oxidation pathway in *E. coli*, different medium-chain α, ω-dicarboxylic acids could be produced from the waste food oil. The engineered strain could produce 15.26 g·L$^{-1}$ medium-chain α, ω-dicarboxylic acids using waste food oil as the sole carbon source after adaptive evolution [61].

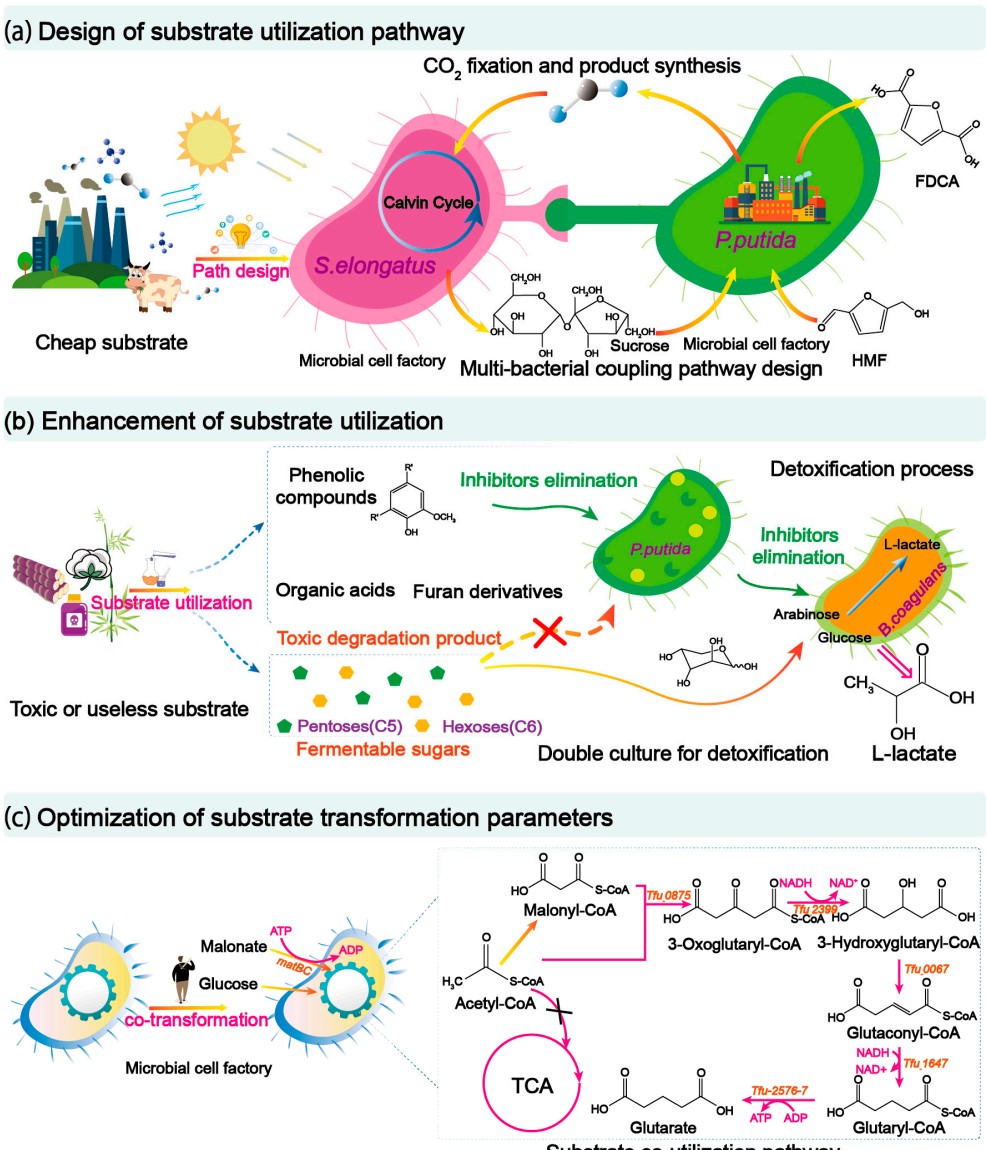

**Figure 2.** Strategies for efficient utilization of cheap substrates. (**a**) A multi-bacterial coupling pathway was designed for the utilization of substrates. *S. elongatus* was used for the fixation of $CO_2$ and the supply of sucrose, while *P. putida* used sucrose to synthesize 2,5-Furandicarboxylic acid to achieve the utilization of substrates. HMF, 5-hydroxymethylfurfural; FDCA, 2,5-Furandicarboxylic acid. (**b**) The toxic substrate is utilized through the two-bacterium co-culture system, and *P. putida* metabolizes the toxic substances in the hydrolysate, promoting the efficient synthesis of L-lactate by *B. coagulans*, to achieve the detoxification treatment of the toxic substrate and microbial synthesis. (**c**) By designing the co-utilization of malonate and glucose, the efficient synthesis of glutarate can be realized.

### 2.1.3. Combining Endogenous and Artificial Substrate Utilization Pathways

Although most of the substrates could be utilized through native or artificial pathways, it is challenging to utilize some substrates by a single microorganism or pathway.

Therefore, such substrates can only be converted into desired bio-based monomers by combining natural pathways and artificial pathways. For example, although there are some carboxylases for carbon fixation in *E. coli*, their efficiencies are quite low [63]. To solve this problem, self-assembled cadmium sulfide nanoparticles were developed to improve the $CO_2$ fixation process and a $CO_2$ mitigation switch was designed to strengthen the $CO_2$ mitigation process. By integrating synthetic $CO_2$ fixation and $CO_2$ mitigation modules, the engineered *E. coli* could produce malate with a yield of 1.48 mol·mol$^{-1}$ glucose by $CO_2$ sequestration [62]. No natural microorganisms can directly use sugar to produce 1,3-PDO. To this end, a de novo 1,3-PDO biosynthesis route from sugar was designed by expanding the homoserine synthesis pathway. Combined with protein engineering and the expression of native alcohol dehydrogenase, the engineered *E. coli* could directly produce 1,3-PDO from the sugar substrate [64].

### 2.1.4. Building Orthogonal Substrate Utilization Pathways

A common problem during microbial production is that carbon flux leaks into pathways other than the target pathway, which might decrease the yield of desired chemicals. To solve this problem, a strategy termed parallel metabolic pathway engineering was proposed in which one kind of carbon source was designed for desired chemical biosynthesis, while another type of carbon source was engineered for supporting cell growth [72]. For example, the metabolic pathways of *E. coli* were engineered, so that only glucose could be used for cis, cis-muconic acid production, while xylose was designed for restoring cell growth from the Dahms pathway. As a result, the engineered *E. coli* could produce 4.09 g·L$^{-1}$ cis, cis-muconic acid with a high yield of 0.31 g·g$^{-1}$ glucose, by co-utilizing glucose-xylose mixtures [72]. Similarly, the production yield of 1,3-PDO from glycerol was always limited by the formation of byproducts. To solve this problem, the endogenous glycerol assimilation pathway was eliminated, and mannitol was fed as a co-substrate to improve glycerol flux towards 1,3-PDO synthesis. As a result, glycerol was redirected mainly for 1,3-PDO biosynthesis and the final engineered strain could obtain a 1,3-PDO yield of 0.76 mol·mol$^{-1}$ glycerol [65].

### 2.1.5. Coupling Multi-Bacterial Substrate Utilization Pathways

The introduction of a heterogeneous metabolic pathway into a strain often causes cell burden, which affects cell growth performance as well as the synthetic efficiency of material monomers [115]. To solve this issue, a co-culture strategy was adopted in which one strain was used for the degradation of complex substrates, while the other strain was designed for the synthesis of monomers. For example, by co-culturing the cellulolytic fungus *T. reesei* and the lactic acid-producing *R. delemar*, lactic acid was produced from microcrystalline cellulose using a fungal consortium [74]. Synthetic consortia can exhibit advantages compared to pure strain because they can combine the functions of different strains to achieve the direct utilization of some special substrates, such as $CO_2$. For instance, a syntrophic consortium was recently proposed by co-culturing engineered *S. elongatus* and *P. putida*. In this system, $CO_2$ can be fixed for producing sucrose by *S. elongates*, and the produced sucrose could support the growth of *P. putida*, catalyzing the conversion of 5-Hydroxymethylfurfural to 2,5-Furandicarboxylic acid (Figure 2a) [82].

### 2.2. Enhancing Substrate Utilization Capacity

Designing various metabolic pathways for the utilization of cheap and abundant substrates into high-value-added compounds is necessary. However, not all substrates can be directly utilized without pretreatment [116,117]. Therefore, it is particularly important to strengthen the utilization capacity of the substrates. To achieve this goal, currently, a total of five main strategies could be summarized: (i) expanding substrate spectrum; (ii) optimizing mass transfer; (iii) depolymerizing substrates; (iv) detoxifying substrates; and (v) enhancing strain growth performance.

### 2.2.1. Expanding Substrate Spectrum

In addition to traditional five-carbon and six-carbon substrates, some polymeric sugars/substrates are difficult to be utilized by microorganisms [66,118]. By exploiting technologies for expanding the substrate spectrum in the biotransformation process, the overall production cost could be greatly decreased. For example, the wild-type *C. glutamicum* strain cannot utilize xylooligosaccharide as the substrate for chemical production. To solve this issue, a cell surface display technique was proposed in which β-xylosidase was expressed on the *C. glutamicum* cell surface using the PorH anchor protein. Furthermore, the xylose assimilation pathway was also introduced to enable the engineered *C. glutamicum* to produce 11.6 mM cadaverine using xylooligosaccharide as substrate [50]. With the same technology, α-amylase was expressed on the cell surface of *E. coli* to enable the direct production of 1,2-propanediol (1,2-PDO) and 1,3-PDO from starch [66].

### 2.2.2. Optimizing Mass Transfer

The improvement of microbial substrate utilization ability is often accompanied by the improvement of substrate mass transfer efficiency. To increase the substrate mass transfer efficiency, two main methods could be carried out, including (i) developing a water-organic conversion system to enhance interfacial mass transfer efficiency; and (ii) developing biofilm technology to expand substrate enzyme contact area. For example, paraxylene (PX) and terephthalic acid (TPA) are volatile and almost insoluble in water. To increase the mass transfer efficiency, oleyl alcohol with biocompatibility was recruited as an organic phase for biphasic microbial transformation. As a result, PX can be dissolved in oleyl alcohol and allowed to partition into the aqueous phase with a special partition coefficient. In this biphasic microbial transformation system, 6.9 g·L$^{-1}$ of TPA could be produced after 46 h bioconversion [83]. As extracellular organelles, biofilms can be formed on the cell surface to promote the self-assembly of enzymes or nanoparticles to improve the utilization of substrates [119–122]. To enable the utilization of starch in *E. coli*, a light-driven CdS-biohybrid system was developed to increase the contact surface area of microbial cell membranes as well as improve the intracellular NADH concentration [123].

### 2.2.3. Depolymerizing Substrates

At present, various strategies for utilizing natural or unnatural substrates have been developed. However, most of these methods are not applicable for substrates with highly polymerized substrates, such as biomass feedstock and polymeric materials. To degrade such substrates, different depolymerization methods, including chemical solvent treatment and physical crushing methods, have been developed [124]. For example, an aqueous solvent (NaOH/ChCl:TH/water) was recently designed to efficiently hydrolyze sugarcane bagasse and reduce the degree of polymerization of sugarcane bagasse under mild conditions. After pretreatment, the hydrolysates of sugarcane bagasse could be directly used for microbial adipic acid production with a yield of 0.39 g·g$^{-1}$ glucose, providing a cost-effective benefit for adipic acid production [57]. Similarly, a chemo–microbial hybrid process was carried out in *E. coli* to convert PET to 2-pyrone-4,6-dicarboxylic acid (PDC) with a yield of up to 96%. Specifically, PET was firstly depolymerized to TPA via microwave-assisted hydrolysis using biomass-derived SiO$_2$ catalysts with thiol functionalization. Then, the produced TPA was used as the substrate for PDC production with a whole-cell conversion strategy [84].

### 2.2.4. Detoxification of Substrates

Substrate detoxification is also helpful to improve the substrate utilization capacity. Lignocellulosic biomass is an attractive and sustainable alternative to petroleum-based feedstock for chemical production. However, it usually needs to be pretreated before they can be efficiently used for the biosynthesis of monomers. During the pretreatment process, a variety of inhibitors were inevitably produced that seriously hinder the growth of microorganisms [125]. To solve this problem, a synthetic consortium could be developed

to achieve the detoxification treatment of inhibitors. For example, in a synthetic consortium of engineered *P. putida* and *B. coagulans*, *P. putida* was designed to detoxify toxic substrate in lignocellulosic hydrolysate, while *B. coagulans* was used to capture carbon sources from the hydrolysate to produce 35.8 $g \cdot L^{-1}$ lactic acid (Figure 2b) [75]. A similar study was conducted by building a *P. putida* and *E. coli* consisting synthetic consortium. As a result, 1.02 $g \cdot L^{-1}$ medium-chain polyhydroxyalkanoate was produced using lignocellulosic hydrolysate as the substrate [85].

### 2.2.5. Enhancing Strain Growth Performance

Various factors, such as high concentrations of substrate, substrate toxicity, and environmental stress, can result in insufficient cell growth and decreased production performance. To this end, microbial cell growth could be enhanced by fermentation optimization and cell immobilization. For example, by optimizing the culture conditions, the biomass of *C. butyricum* L4 could be improved to increase the 1,3-PDO titer to 70.1 $g \cdot L^{-1}$ [67]. Free cells are often affected by complex fermentation environments and cannot be reused, thus increasing production costs. Cell immobilization based on biofilm is a promising strategy to maintain cell activity during fermentation production. For instance, a glycosylated membrane with rhamnose modified surface was constructed to immobilize *A. succinogenes* for producing succinic acid. With a reduced cell lag stage and high reusability, the biofilm-based cell-immobilized fermentation technology enables the engineered strain with an 18% increased succinic acid titer compared with that of free cell fermentation [86]. In addition, the design and modification of the bioreactor to better maintain the stable culture medium resulted in better growth and production of microbial cells, and the concentration of 1,3-PDO was increased to 88.6 $g \cdot L^{-1}$ [126].

### 2.3. Optimizing Substrate Conversion Process

In addition to the design of substrate utilization pathways and enhancement of substrate utilization capacity, the optimization of the substrate conversion process can also be improved to promote the efficient absorption and utilization of substrates. In general, five strategies could be put forward, such as (i) optimizing substrate transport by overexpressing a specific transporter; (ii) relieving carbon catabolite repression to steadily improve substrate absorption; (iii) co-utilizing multi-substrate to couple cell growth and production; (iv) achieving one-step biotransformation to simplify the substrate utilization process; and (v) strengthening energy supplement to build efficient substrate transport.

### 2.3.1. Optimizing Substrate Transport

The utilization of substrate can be affected by the efficiency of substrate transport. Therefore, it is particularly important to optimize the substrate transport process. Appropriate overexpression of proteins that transport substrate can enhance the absorption of substrates from extracellular to intracellular. For example, by co-expression of $CO_2$ transport and fixation genes of *E. coli* using different promoters, the engineered strain could produce 89.4 $g \cdot L^{-1}$ succinic acid by improving the $CO_2$ fixation process [87]. Similarly, by overexpressing glycerol kinase GUT1 in yeast *Y. lipolytica*, the uptake of glycerol was increased and exhibited a positive effect on cell growth and product synthesis [99].

### 2.3.2. Relieving Carbon Catabolite Repression

Overexpressing substrate transporter can greatly improve the efficiency of the substrate conversion process. However, when multiple substrates, such as glucose and xylose exist together, carbon catabolite repression may occur, resulting in the inhibition of xylose utilization by glucose [127]. To solve this issue, three main approaches could be recruited to alleviate the catabolite repression phenomenon. Firstly, deleting carbon repression control (Crc) genes. For instance, the *Pseudomonas putida* could be engineered to convert ferulate to muconate. However, the existence of glucose, which was used for supplementary energy and cell growth in the medium, always causes carbon catabolite repression, leading to

impaired production performance. To this end, the Crc encoding gene was deleted and the resulting strain could produce muconate from ferulate with a doubled yield [128]. Secondly, screening xylose catabolic operon XylR mutant. For example, by introducing R121C and P363S into XylR, the xylose catabolism was found to be activated without catabolite repression control. As a result, the engineered *E. coli* strain exhibited a 50% increase in lactate titer than that of the wild-type strain using a glucose-xylose mixture substrate [76]. Thirdly, Knocking out glucose transporter Crr. Due to the metabolite repression of glucose on glycerol, the glucose transporter Crr of *K. pneumoniae* was knocked out to increase the equivalent of glycerol into the synthesis pathway of 1,3-PDO. The engineered strain exhibited an increased 1,3-PDO titer from 61 to 78 $g \cdot L^{-1}$, with a glycerol conversion rate reaching 59.5% [68].

### 2.3.3. Co-Utilizing Multi-Substrate

Glucose was mainly used for cell growth and production, while some substrates could strengthen the intracellular reductive force and act as a synthetic precursor. To achieve the co-utilizing of multi-substrate, different substrate utilization pathways can be introduced in one strain to coordinate the co-utilization of substrates. For example, by building glycerol and glucose substrate utilization pathways, the NADH flux and metabolic flux for the production of succinic acid could be improved, as the conversion of glycerol to PEP generates twice as much reducing equivalents mole compared with glucose [88]. To obtain more precursors for the synthesis of bio-based monomer, the utilization pathway of malonic acid was introduced into a glutaric acid-producing *E. coli.* The resulting strain produced more precursor malonyl-CoA compared to the control strain, increasing the glutaric acid titer to 6.3 $g \cdot L^{-1}$ using both glucose and malonic acid as substrates (Figure 2c) [100].

### 2.3.4. Achieving One-Step Biotransformation

To utilize polymeric substrates, such as starch, cellulose, and xylan, multiple independent biological processes are needed, including the production of hydrolases, hydrolysis of substrates to monosaccharide, and metabolism of monosaccharide to desired products. To decrease overall production costs and simplify the process, one-step biotransformation could be designed to enable the substrate to be saccharified and fermented at the same time [129]. For example, by adding glucoamylase into the fermentation system, the liquefied cassava starch was directly used for the production of succinic acid by *E. coli* NZN111 [130]. Commercial enzymes are expensive and their addition for saccharification is not economically viable for the large-scale fermentation process. To solve this problem, some specific strains can be recruited for simultaneous saccharification and fermentation of the substrate due to their hydrolyzing and fermenting capabilities, thereby reducing production costs. For instance, an isolated *L. manihotivorans* DSM 13343 was used for producing 18.69 $g \cdot L^{-1}$ lactic acid from a mixed culture with food waste as substrate [77].

### 2.3.5. Strengthening Energy Supplement

In addition, the optimization of energy supply in the biotransformation process enables microorganisms to utilize substrates with reduced energy consumption, increasing the utilization efficiency of substrates. To achieve this goal, the intracellular ATP concentration can be enhanced to compensate for substrate absorption and transport processes that consume energy, such as carbon fixation and xylose transport. For example, by overexpressing the ATP-generating phosphoenolpyruvate carboxykinase combined with the $CO_2$ fixation pathway, the $CO_2$ fixation efficiency for the malic acid biosynthesis pathway was increased by 110% [63]. Similarly, the ATP-generating phosphoenolpyruvate carboxykinase could also be co-expressed with the xylose utilization pathway, increasing the titer of succinic acid to 11.13 $g \cdot L^{-1}$ using corn straw as substrate [89].

Through the design of the substrate utilization pathways, the enhancement of utilization capacity, and the optimization of the conversion process, substrate utilization plays an important role in bio-based monomer biosynthesis. The advantages of developing efficient

substrate utilization strategies can be summarized in three aspects: (i) Reduce production costs. To improve the market competitiveness of bio-based monomers compared with petrochemical-based monomers, it is very important to develop strategies for efficient utilization of cheap and abundant substrates; (ii) Environmental protection. Domestic garbage often causes resource waste and environmental pollution, and the development of microbial utilization strategies is the key to achieving green biological manufacturing [61]; (iii) Microbial synthesis performance improvement. The implementation of a series of strategies, including building orthogonal pathways, relieving carbon catabolite repression, strengthening energy supplements, etc., can enhance the efficiency of engineered strains in bio-based monomer production.

## 3. Improving Bio-Monomer Synthetic Efficiency

Through the efficient utilization of cheap substrates, the production cost of microbial bio-based materials can be greatly decreased. In addition, to promote the conversion of substrate into target products effectively, the synthetic efficiency of the microbial cell factories can be improved from different regulatory levels, such as enzymes, pathways, and the engineered cell itself. At present, there are three methods to improve the synthetic efficiency of monomers, which include: (i) strengthening key enzyme performance; (ii) optimizing synthetic pathway efficiency; and (iii) regulating cellular metabolic networks (Figure 3).

### *3.1. Strengthening Key Enzymes Performance*

It is a common strategy to improve the production efficiency of monomer microbial cell factories by strengthening the performance of key enzymes. The performance of pathway enzymes could be improved from five aspects: (i) screening heterologous enzymes to obtain more suitable enzymes; (ii) promoting protein correct folding to maximize the enzyme activity; (iii) enhancing protein expression level to drive more metabolic flux; (iv) enzyme directed evolution to modify enzyme properties; and (iii) establishing enzyme recycling technology to reduce the enzymes demand.

### 3.1.1. Screening Heterologous Enzymes

The performance of pathway enzymes often plays a crucial role in determining the synthetic efficiency and reaction direction in a synthetic pathway. In the microbial synthesis pathway, when the activity of endogenous enzymes is not high enough or there is a lack of enzymes that can play a specific role, it can be optimized by screening for heterologous enzymes with high catalytic efficiency or cofactorless enzymes that can help alleviate the cofactor restriction of endogenous enzymes. For example, malate dehydrogenase (MDH) from different sources was screened and characterized in the engineered *M. succinicipro-ducens*. As a result, the MDH from *C. glutamicum* exhibited the highest enzyme activity and was further used to replace endogenous MDH in the engineered *M. succiniciproducens*, resulting in succinic acid titer reaching 134.25 g·L$^{-1}$ [88]. In a similar study, the enoate reductase of *B. coagulans* was introduced into *S. cerevisiae*. With a three-stage fermentation process optimization, the final engineered strain could directly produce adipic acid using glucose as substrate [58].

### 3.1.2. Promoting Enzyme Folding

The correct folding of enzymes also plays a positive effect on the enhancement of enzyme properties to a certain extent. With correct protein folding, the expression level as well as the catalytic activity of pathway enzymes could be improved to speed up the biosynthesis process. Usually, protein folding can be tuned by optimizing protein expression conditions [103], expression host, expression plasmid, signal peptides, molecular tags, and chaperones [131]. For example, by introducing zwitterionic peptides into lysine decarboxylase to promote correct folding, the enzymatic activity of engineered lysine decarboxylase was double that of wild-type lysine decarboxylase [54]. There are three main

elements of the chaperone systems in *E. coli*, namely, trigger factor, GroEL-GroES, and DnaK-DnaJ-GrpE, which can facilitate the correct folding of polypeptides and prevent the formation of inclusion bodies. By introducing trigger factor chaperone into an engineered E. coli harboring the recombinant 1,2,4-butanetriol pathway, the titer of 1,2,4-butanetriol could be increased from 0.56 g·L$^{-1}$ to 1.01 g·L$^{-1}$ [106].

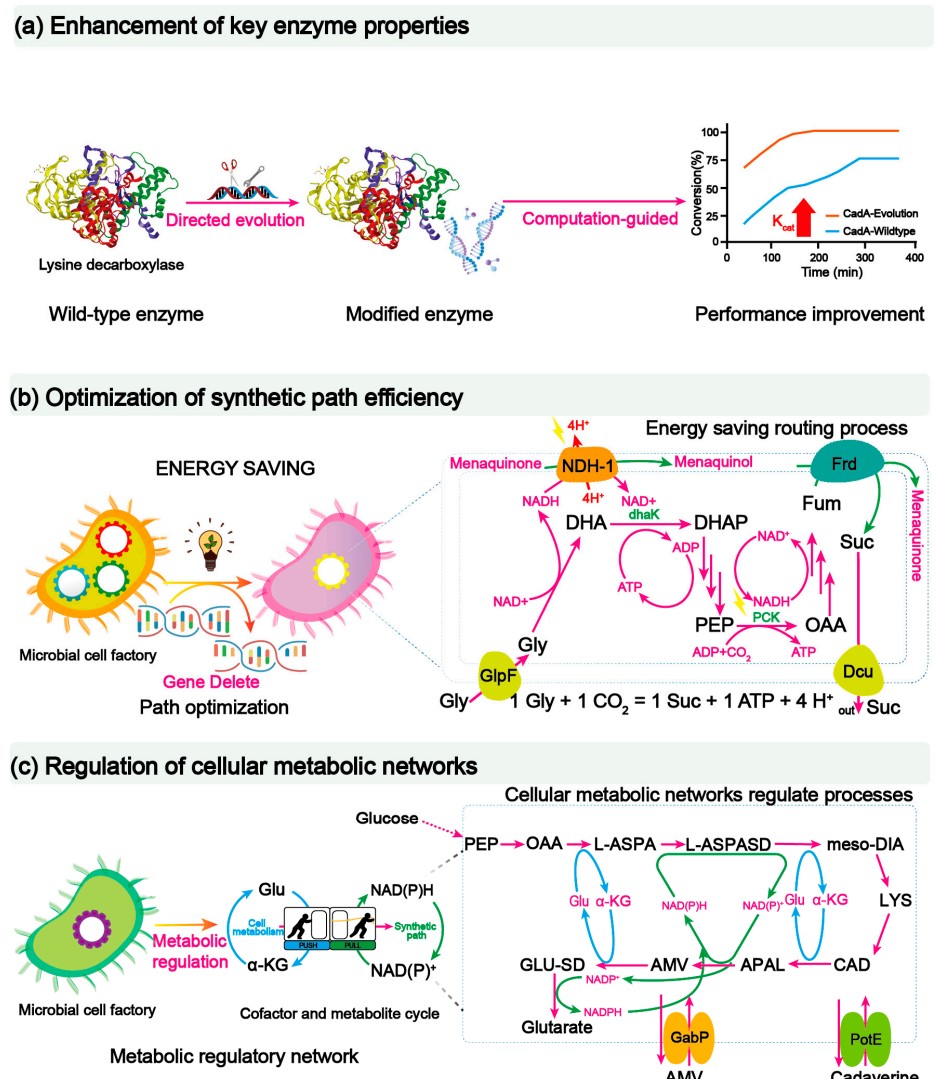

**Figure 3.** Strategies for improving synthesis efficiency. (**a**) Aiming at the problem that lysine decarboxylase is not resistant to an alkaline pH environment, directional evolution is introduced to improve the catalytic efficiency of the process and strengthen the performance of L-lysine decarboxylase. (**b**) The growth of *E. coli* and the synthesis of succinic acid were improved by constructing a glycerol metabolic pathway with low energy requirements, and the efficiency of the synthetic pathway was increased. Abbreviations: DhaK, ATP-dependent dihydroxyacetone kinase; PCK, phosphoenolpyruvate carboxykinase; Frd, fumarate reductase; Dcu, anaerobic C4-dicarboxylate transporter Dcu; NDH-1, NADH: ubiquinone oxidoreductase I; GlpF, glycerol facilitator; DHA, dihydroxyacetone; DHAP, dihydroxyacetone 3-phosphate; PEP, phosphoenolpyruvate; OAA, oxaloacetate; Fum, fumarate; Suc, succinate; Gly, glycerol. (**c**) Based on the metabolic drive design, the precise coupling of the metabolic flow between the cell's central metabolism and the glutarate synthesis pathway was achieved, thus improving the synthesis efficiency of glutarate. Glu, glutamate; α-KG, α-ketoglutarate; PEP, phosphoenolpyruvate; APAL, 5-aminopentanal; AMV, 5-aminovalerate; OAA, oxaloacetate; L-ASPA, L-Aspartate; L-ASPASD, L-Aspartate semialdehyde; meso-DIA, meso-Diaminopimelate; CAD, cadaverine; LYS, lysine; GLU-SD, glutarate semialdehyde.

### 3.1.3. Enhancing Protein Expression Level

Enhancing protein expression levels is a simple way to strengthen metabolic efficiency, remove metabolic bottlenecks, and regulate metabolite distribution. Many traditional expression regulatory elements can be used, such as using stronger promoters and high-copy plasmids. For example, to reduce the accumulation and inhibition of cadaverine in engineered *E. coli* cells, a bi-directional cadaverine transporter PotE was overexpressed using high-copy plasmid to drive more metabolic flux from substrate to cadaverine and the desired product, glutaric acid [101]. In enhancing protein expression levels, some species are more suited for genomic modification than others.

### 3.1.4. Enzymes-Directed Evolution

Pathway enzyme activity could be easily affected by the external as well as the internal environments; thus, it is necessary to modify enzymatic properties, such as catalytic activity, catalytic stability, and stereoselectivity, to improve microbial cell performance. To do so, the simplest and most effective method is to carry out protein-directed evolution [132,133]. For example, lysine decarboxylase plays a key role in the process of converting lysine decarboxylation to cadaverine, but it is often affected by the alkaline product and the reaction temperature, which hinders its application in industrial production. To this end, the pH stability of lysine decarboxylase was engineered with protein-directed evolution. The resulting mutant M3 exhibited a 6-fold increase in cadaverine production at pH 10.0 than that of the control strain. More importantly, the engineered strain harboring this mutant M3 could produce $418 \text{ g·L}^{-1}$ cadaverine in 15 h, which is the highest titer produced to date [51]. Different enzyme properties can be improved simultaneously using protein-directed evolution. For example, both the thermal and alkaline stability of lysine decarboxylase could be improved by combining directed evolution and computation-guided virtual screening. The engineered strain with the best mutant could produce $160.7 \text{ g·L}^{-1}$ cadaverine at $50\,^\circ\text{C}$ without pH regulation (Figure 3a) [52].

### 3.1.5. Establishing Enzymes Recycling

Some bio-based monomers, such as glutaric acid and cadaverine, were usually produced by the whole-cell transformation process to obtain high productivity. However, the production cost depends on the input of each batch of enzymes or cells. To solve this problem, various enzyme or cell immobilization strategies could be proposed and exhibited advantages, such as (i) reducing the process cost by recycling cells or enzymes; and (ii) maintaining the catalytic activity of enzymes to resist external environmental stress [102]. For example, in a recent study, chitin biopolymer was recruited as a functional material to mediate the immobilization of pyridoxal 5′-phosphate and lysine decarboxylase. Under this design, a continuous biosynthesis of cadaverine without the exogenous addition of pyridoxal 5′-phosphate was achieved [134]. A similar study was conducted for glutaric acid biosynthesis. Specifically, a microbial consortium was constructed by immobilizing two types of engineered *E. coli* cells with colloidal chitin. This efficient microbial platform exhibited good stability and repeatability, resulting in $73.2 \text{ g·L}^{-1}$ glutaric acid accumulation from lysine [102].

### 3.2. Optimizing Synthetic Pathway Efficiency

In addition to the strengthening performance of key enzymes, optimizing the synthetic pathway can also improve the efficiency of monomer biosynthesis to a certain extent. The microbial monomer synthetic pathways are often accompanied by many factors, such as long metabolic reaction steps, by-product accumulation, high energy consumption, etc. Therefore, it is expected that the substrate can be converted into the target products more efficiently by optimizing the synthetic pathway efficiency. At present, five main strategies have been developed, including (i) balancing enzyme expression level; (ii) reconstructing target metabolic flux; (iii) reducing pathway energy consumption; (iv) shortening the spatial distance; and (v) dynamic pathway regulation.

### 3.2.1. Balancing Enzyme Expression Level

For multiple-enzyme-involved synthetic pathways, metabolic bottlenecks occur frequently because of the differences in catalytic performances and expression levels of pathway enzymes. Such metabolic bottlenecks will lead to the accumulation of intermediate metabolites and further reduce the catalytic efficiency of the metabolic pathway. To this end, the expression level of each enzyme can be regulated to improve the pathway efficiency. Various strategies have been developed to balance enzyme expression levels, such as plasmid optimization [102], promoter engineering [135], and ribosome binding site engineering [136]. For example, a total of seven enzymes involved in the glutaric acid biosynthetic pathway were arranged into plasmids with different copy numbers for combinatorial optimization [102]. These pathway enzymes can be further optimized with different strengths of promoters and ribosome binding sites. The final engineered *E. coli* strain could produce 77.62 g·L$^{-1}$ glutaric acid with lysine substrate [103].

### 3.2.2. Redirecting Target Metabolic Flux

Microorganisms have evolved extensive regulation and complex interactions between metabolic pathways. As a result, simply introducing a heterologous synthetic pathway for chemical production is often inefficient due to metabolic fugitive effects. Two main approaches can be implemented to redirect the metabolic flux toward target product synthesis. Firstly, increasing precursor concentrations by overexpressing key enzymes in a synthetic pathway. For example, by optimizing the malonyl-coA synthetic pathway to improve the supply of precursors, the malonic acid titer could be increased to 1.62 g·L$^{-1}$ in *S. cerevisiae* [107]. Similarly, by enhancing the accumulation of 5-aminovaleric acid, the titer of glutaric acid reached 22.7 g·L$^{-1}$ in the engineered *C. glutamicum* [40]. Secondly, reducing the metabolic loss in the branched pathway by blocking the by-product synthetic pathways. For instance, in a recent study, an autonomous circuit, containing CRISPRi, stationary phase promoter, and protein degradation tag, was successfully constructed for rewiring metabolic flux. By targeting byproduct synthetic pathways, the concentrations of acetic acid, lactic acid, and formic acid in the glutaric acid-producing strain were decreased to 40%, 41%, and 35% compared to that of the control strain, respectively [104].

### 3.2.3. Reducing Pathway Energy Consumption

A third method to improve the synthetic pathway efficiency is to reduce the energy consumption in the target metabolic pathway. By reducing energy consumption, more carbon flux can be rewired to the synthetic pathway in a thermodynamically advantageous manner. To achieve this goal, two main approaches have been developed. Firstly, replacement of high energy consumption pathway. For example, to promote anaerobic succinic acid production, an ATP-dependent dihydroxyacetone kinase from *Klebsiella* was used to replace the native phosphoenolpyruvate-dependent dihydroxyacetone kinase of *E. coli*. As a result, the engineered *E. coli* strain could save one mole of NADH per glycerol to increase succinic acid titer by 282% (Figure 3b) [90]. A second way to reduce pathway energy demand is to fine-tune hybrid pathways. By phosphoketolase-mediated non-oxidative glycolysis to accommodate the output NAD(P)H/acetyl-CoA stoichiometries to match the energy demand of the 1,3-butanediol synthesis pathway, the yield of 1,3-butanediol reaches 113% of the theoretical maximum from native metabolism [69].

### 3.2.4. Constructing Substrate Channels

Constructing a substrate channel is one of the important methods to reduce the escape of intermediate metabolites and improve the efficiency of multi-enzyme cascade reactions. Various scaffold-based strategies have been designed, including immobilizing different pathway enzymes on the DNA, RNA, or protein scaffolds to shorten their spatial distance and improve the catalytic efficiency of the synthetic pathway [137]. For example, a synthetic protein scaffold was used to shorten the spatial distance between the myo-inositol oxygenase and myo-inositol-1-phosphate synthase, and the resulting engineered *E. coli*

exhibited a 5-fold improvement of glucaric acid production than that of the non-scaffolded control strain [108]. Similarly, by arranging phosphoenolpyruvate carboxylase and malate dehydrogenase on a synthetic protein scaffold complex in the engineered *E. coli*, the malic acid productivity was improved by 3.6-fold [138].

### 3.2.5. Dynamic Pathway Regulation

Although the above static regulation optimization strategies can effectively improve the efficiency of the synthesis pathway, matching the production process and growth process through dynamic control is also an effective means. With dynamic pathway regulation, the carbon flux can be directed toward target product synthesis at a specific time or intermediate metabolite concentration [139,140]. For example, a sensor-regulator and RNAi-based bifunctional dynamic switch was designed to enable a muconic acid dose-dependent upregulation of the target biosynthetic module and downregulation of the competing pathway genes. The resulting engineered *E. coli* strain could produce 1.8 g·L$^{-1}$ muconic acid, which was substantially higher than the static controls [73]. Various dynamic switches can also be designed according to the external environment changes. For example, a low-pH-induced promoter Pgas was recruited to dynamically control the expression of cis-aconitate decarboxylase gene in the production stage, allowing itaconic acid titer to reach 4.92 g·L$^{-1}$ [110].

### 3.3. *Regulating Cellular Metabolic Networks*

The regulation of the cell metabolic network is also an effective strategy to improve monomer synthetic efficiency. Both the intracellular carbon flux and redox balance can be optimized to improve the synthetic efficiency of engineered microorganisms. To realize this goal, five main methods can be proposed, including (i) omics-assisted key targets identifying; (ii) decoupling cell growth and production; (iii) metabolic driving design; (iv) engineering transcription factor; and (v) tuning redox homeostasis.

### 3.3.1. Omics-Assisted Key Targets Identifying

To adapt to the complex and changeable external environment, microorganisms have evolved a variety of feedback and negative feedback regulatory networks. As a result, rational metabolic engineering often fails to build efficient microbial cell factories. To solve this problem, (i) comprehensive omics data, such as genomic, transcriptomic, and fluxomic, as well as (ii) building a screening library, can be performed to identify engineering targets for enhanced biomaterial monomer production. For example, with genomic analysis in *C. glutamicum*, a total of 11 key genes related to glutaric acid biosynthesis were obtained. After combinational engineering of these genes, the resulting strains can produce glutaric acid with a titer of 105.3 g·L$^{-1}$ [105]. To obtain the key genes related to cadaverine production, 67 genes-repressing sRNAs were constructed and tested in *E. coli*. The best anti-serA strains can produce cadaverine with a titer of 13.7 g·L$^{-1}$ [53]. In addition, through laboratory adaptation evolution (ALE) of *P. puticosa* in xylose substrate medium and subsequent genomics analysis, the modification targets of *aroB* gene and *Xyle* mutations were identified, which were favorable for the growth of the strain, and the yield of muconate reached 33 g·L$^{-1}$ [141].

### 3.3.2. Decoupling Cell Growth and Production

The carbon flux competition between cell growth and monomer synthesis always hinders the development of efficient microbial cell factories. When the excess carbon flow enters the biomass form module, the cell biomass will increase, but at the same time, the yield of the monomer will be reduced; on the contrary, if more carbon flux enters the synthetic pathway, the cell biomass will be decreased, resulting low final product titer. To decouple cell growth and production, different dynamic switches can be designed according to the changes in fermentation conditions or cell growth status. For example, an oxygen-dependent dynamic regulation system was established to regulate the expression of

key genes in aerobic and anaerobic stages. The distribution of carbon flux in the engineered *E. coli* can be regulated to increase adipic acid titer by 41.62-fold [59]. Dynamic regulation can be achieved according to different cell statuses. For instance, by integrating growth phase promoters and degrons, a protein abundance bifunctional molecular switch was designed to decouple *E. coli* cell growth from D-glucaric acid biosynthesis, resulting in the production of 1.16 g·L$^{-1}$ glucaric acid [109].

### 3.3.3. Metabolic Driving Design

The cofactors and precursors in microbial cell factories play a crucial role in the biosynthesis of target products. By coupling the regeneration and consumption of cofactors or precursors, the metabolic driving force could be created to strengthen microbial metabolism, thus improving the synthetic efficiency of the target products. For example, by recycling the NADP$^+$ and $\alpha$-ketoglutaric acid generated from the lysine catabolic pathway to an NADP$^+$ and $\alpha$-ketoglutaric acid consumed glutaric acid biosynthesis pathway in engineered *E. coli*, the glutaric acid biosynthetic pathway could be improved to produce 54.5 g·L$^{-1}$ glutaric acid (Figure 3c) [101]. A similar design was carried out in a 2,3-butanediol-producing *E. coli*, in which enforced ATP wasting was introduced to consume ATP. As a result, more flux was redirected to the target synthetic pathway with ATP generation to increase the titer of 2,3-butanediol 10-fold [111].

### 3.3.4. Engineering Transcription Factor

Transcription factors can regulate multiple genes or even global metabolic network genes, so they can be used as important targets to enhance the efficiency of microbial monomer synthesis. For example, by deleting the global transcriptional regulators arcA and glpR in *V. natriegens*, the carbon flux towards the 1,3-PDO synthesis pathway was enhanced by transcriptomics analysis, increasing the yield of 1,3-PDO to 0.50 mol·mol$^{-1}$ glycerol [70].

### 3.3.5. Tuning Redox Homeostasis

At the same time, the REDOX balance of the microbial cell factory can also be engineered to improve monomer synthetic efficiency. Redox homeostasis can be achieved by tuning either the ATP or NAD$^+$ concentrations. For example, by adjusting intracellular ATP concentration [90] and NADH/NAD$^+$ concentration ratio [142] to meet the requirement of succinic acid biosynthesis, the titer of succinic acid can be increased by 282% and 39%, respectively, showing good potential in engineering microbial cell factory.

After strengthening the performance of key enzymes, optimizing the efficiency of the synthetic pathway, and regulating the cell metabolic network, the efficiency of the microbial cell factory can be significantly improved. Combined with the efficient utilization of cheap substrates, the prospect of monomer microbial biosynthesis has made an indelible contribution to the development of green biological manufacturing. In the future, by integrating logic gates and sensitive biosensor genetic parts, the optimal level of enzymes would be expressed at the correct time to improve production efficiency [143,144].

## 4. Strengthening Cell Environmental Tolerance

With the development of synthetic biology, the osmotic stress and metabolite stress caused by the accumulation of target products, as well as the acid or base changes caused by the products, make microbial cells more vulnerable to damage or inhibition, thus affecting cell growth and production [143]. Therefore, it is critically important to enhance microbial tolerance to environmental stress. At present, the tolerance of chassis cells can be mainly strengthened in three aspects: (i) enhancing acid-base stress tolerance; (ii) improving osmotic stress tolerance; and (iii) enhancing metabolite stress tolerance (Figure 4).

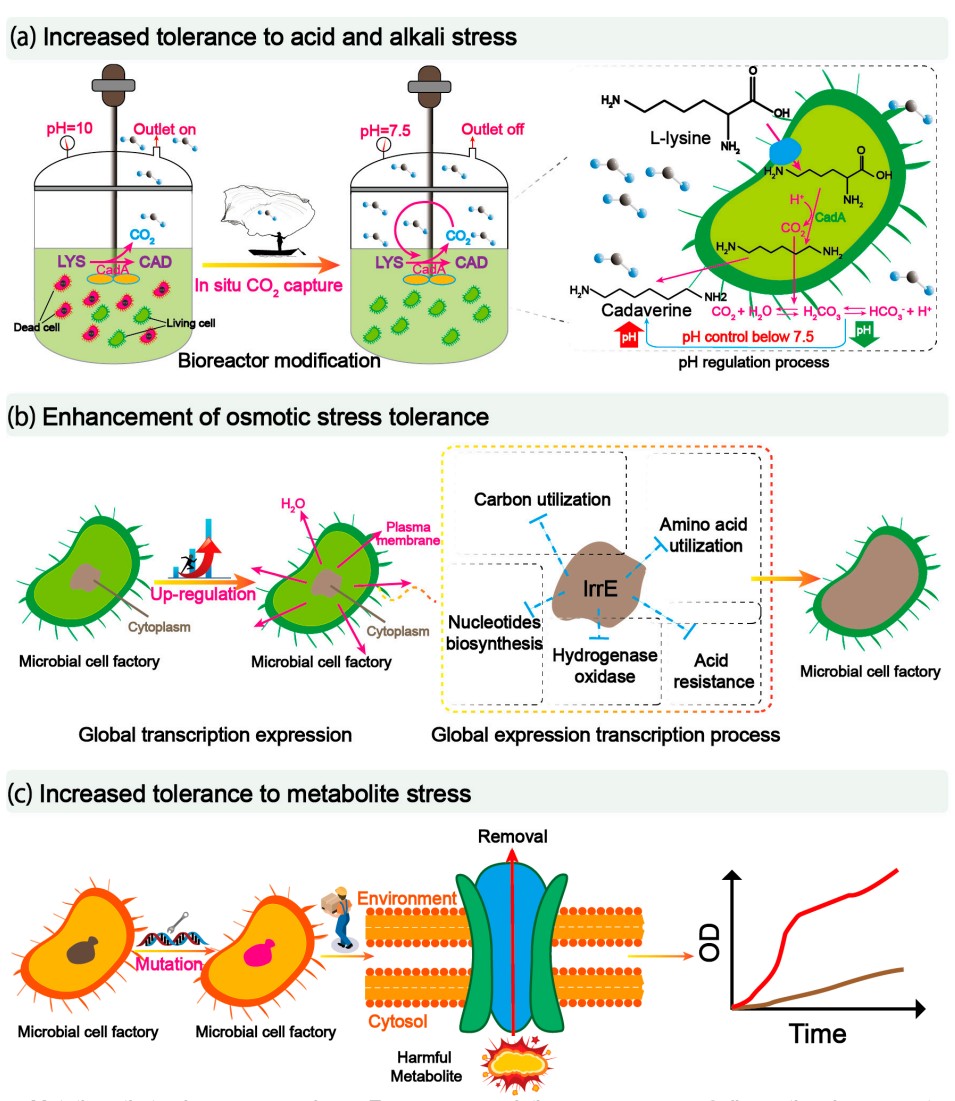

**Figure 4.** Strategies for enhancing chassis cell environmental adaptation. (**a**) The in situ $CO_2$ capture strategy was used to dissolve free $CO_2$ to reduce the pH value of the conversion synthesis system of cadaverine, thus strengthening the acid-base tolerance stress of chassis cells. LYS, L-lysine; CadA, L-lysine decarboxylase; CAD, cadaverine. (**b**) Express the global transcription factor IrrE, improve the synthesis of *E. coli* osmoprotector and osmotic pressure tolerance, enhance cell growth and butyric acid synthesis under hypertonic environment, and achieve the enhancement of osmotic tolerance stress of chassis cells. (**c**) Overexpression of multidrug tolerance transporter QDR3 improved the tolerance of yeast cells to adipic acid and enhanced the tolerance of chassis cells to metabolite stress.

## 4.1. Enhancing Acid-Base Stress Tolerance

The pH values of the fermentation system will change along with the accumulation of products. As acidic products accumulate, acid ions increase in the fermentation system, resulting in a decrease in the pH of the fermentation system, such as succinate. Therefore, it is necessary to improve cell acid-base tolerance to reduce the side effect of the fermentation system on microbial cells. Currently, these strategies can be summarized into four categories: (i) introducing protective agents; (ii) cell membrane engineering; (iii) expression of the acid-resistant gene; and (iv) screening of resistant strains.

### 4.1.1. Introducing Protective Agents

A common regulatory strategy to maintain cell homeostasis is to introduce protective agents, such as pH neutralizers or antioxidants, to maintain the steady state of the fermenta-

tion system. For example, to prevent the pH from increasing during cadaverine production using lysine as the substrate, the bioreactor was modified to recapture the $CO_2$ released in the lysine decarboxylation process. By recruiting $CO_2$ as a pH neutralizer and re-dissolving it into the fermentation system, the pH value of the fermentation system can be reduced to enable a conversion yield of 0.99 mol·mol$^{-1}$ lysine (Figure 4a) [55]. A similar study was conducted in the 1,3-PDO production. By establishing in situ $CO_2$ recapture technology, the pH value of the system can be stable to enable the concentration of 1,3-PDO to reach 88.6 g·L$^{-1}$ [126]. A second way to maintain cell homeostasis is to add antioxidants to the fermentation system. For instance, glutathione can be added to the fermentation system to improve the intracellular pH of *Lactococcus lactis*, improving acid stress resistance [79].

### 4.1.2. Cell Membrane Engineering

Although adding protective agents can promote the growth and production of microorganisms, it also increases the overall cost of monomer production. Therefore, engineering microbial cell factories to improve system homeostasis has received more attention. As the first barrier against environmental stress, the cell membrane can be engineered to enhance cell environmental resistance [145]. For example, by increasing the content of trans-unsaturated fatty acid to change the fluidity of the cell membrane, the cell tolerance of *M. succiniciproducens* can be improved to enable an increased succinic acid production compared with the control strain without membrane engineering [98]. Despite changing the membrane components, membrane transporter can be engineered for improving acid or base stress tolerance. For instance, the overexpression of membrane transporter ZitP and ZitQ in *L. lactis* was proven to have a significant effect on improving cell acid tolerance [146].

### 4.1.3. Expression of the Acid-Resistant Gene

In addition, it is also an effective strategy to engineer genes that can improve cell survival under tolerance stress. These targets included genes that can improve proton conversion in cell metabolism or some acid tolerance genes. For example, by overexpressing hydrogenase accessory proteins HypB-HypC to catalyze proton reduction to produce hydrogen [147], or by introducing three proton-consuming inducible acid resistance systems [148], the engineered cells could survive in extremely low pH conditions. Some genes could be screened for improving cell acid tolerance. For example, by overexpressing the acid-tolerance gene *IoGAS1* in *S. cerevisiae* [80] or the acid-tolerance gene *RecT* in *E. coli* [81], both engineered strains showed good cell growth and productivity under high concentrations of lactic acid.

### 4.1.4. Screening of Resistant Strains

In addition to the above rational metabolic engineering methods, some irrational strain screening methods can also be applied to enhance strain tolerance. ALE is an attractive strategy to obtain resistant strains under different stress conditions. For example, an ammonia-tolerant strain was screened under high ammonia conditions through ALE, as a result, the titer of succinic acid increased to 27.33 g·L$^{-1}$, which was 0.81-fold higher than that of the starting strain *E. coli* BER208 [91].

### 4.2. Improving Osmotic Stress Tolerance

The osmotic stress would increase along with the accumulation of target products in microbial bioproduction, causing a decrease in cell metabolic activity and even cell death. Therefore, it is very important to improve the osmotic stress tolerance of cells to make sustainable cell growth and production. Similar to the method of enhancing acid-base stress tolerance, a total of four strategies could be summarized as follows: (i) strengthening efflux system; (ii) mutation of key proteins and strains; (iii) strengthening global regulation; and (iv) adding protective agent.

### 4.2.1. Strengthening Efflux System

High osmotic pressure in the fermenter is one of the main problems in improving the production efficiency of bio-based monomers. To obtain an efficient microbial cell factory, the efflux system could be strengthened in the engineered strain to improve strain osmotic stress tolerance. For example, during the succinic acid bioproduction process, high concentrations of Cu(I) accumulated in the cell periplasm, causing high osmotic stress. To solve this problem, a Cus copper efflux system was identified to activate CusCFBA expression to transport Cu(I) out of cells to alleviate toxicity. With the introduction of the CusS mutation, the succinic acid-producing strain exhibited a 36% increase in biomass when using a medium supplemented with 30 g·L$^{-1}$ disodium succinic acid [92].

### 4.2.2. Mutation of Key Proteins and Strains

A second method to improve osmotic stress tolerance is the mutation of key proteins and hosts. Some key proteins were proven to play an irreplaceable role in improving osmotic stress tolerance. Thus, they can be rationally regulated to obtain the desired phenotype. For example, by introducing mutations in key stress tolerance genes of DNA-dependent RNA polymerase RpoB, both cell growth and succinic acid production can be increased by more than 40% [93]. Similarly, by screening the hetero-molecular chaperone protein Dnak mutant, the stress tolerance and lactic acid production of *L. lactis* can also be greatly improved [78]. Irrational strain screening can also be implemented to improve microbial osmotic tolerance. For example, succinic acid production was increased 3.3-fold after strain screening through combined mutagenesis (ARTP and $^{60}$Co-γ irradiation) [136].

### 4.2.3. Strengthening Global Regulation

The regulation of osmotic pressure involves the co-expression of multiple genes, so it is difficult to tune a single gene to achieve the change of cell phenotype. To this end, global regulation can be used to obtain osmotic pressure-tolerant strains. Global transcription factors can affect multiple genes and enzyme expressions involved in hyperosmolality, and thus could be selected as the targets to regulate microbial osmotic stress. For example, the global transcription factor cAMP receptor protein CRP can be engineered to increase *E. coli* growth ability by nearly 50% in response to osmotic stress [112]. Moreover, by overexpressing *E. coli* global regulator IrrE, the endogenous synthesis of osmoprotectants (such as trehalose and glycerol) was improved to enable the engineered *E. coli* to produce 24.5 g·L$^{-1}$ succinic acid with 100% seawater medium (Figure 4b) [94].

### 4.2.4. Adding Protective Agent

The last method of improving cell osmotic stress tolerance is to add protective agents to improve enzyme activities. At the late fermentation stage, the specific activities of some pathway enzymes declined gradually owing to the increase of osmotic stress and accumulation of toxic components in the fermentation system. To solve this problem, some protective agents can be added to the medium. For instance, in the later stages of fermentation, all the key enzymes in the succinic acid metabolic network exhibited higher specific activities after the addition of proline. As a result, the osmotic stress tolerance of cells was improved to increase the titer of succinic acid by 22% [95]. In addition, microbial strains can also be engineered so that they can rapidly accumulate protective mediators intracellularly and thus improve cell osmoprotection.

### 4.3. Enhancing Metabolite Stress Tolerance

Metabolite stress caused by the accumulation of metabolites in the later fermentation stage is one of the most common ways to inhibit microbial production. As a result, the target product was rapidly accumulated in the early stage, while accumulating very slowly in the middle and late stages, reducing the overall production efficiency. To solve this problem, different methods have been developed to improve microbial metabolite stress tolerance,

such as (i) adaptive laboratory evolution; (ii) establishing cell mutagenesis; (iii) expressing transporters; and (iv) in situ product separation.

### 4.3.1. Adaptive Laboratory Evolution

ALE is a common method to obtain strains with improved cell metabolite stress tolerance. By gradually increasing the evolutionary stress by enchanting the concentration of the product, evolved strain with target phenotype can be quickly obtained, assisting with a high-throughput screening method. For example, an *S. cerevisiae* with high lactate tolerance was obtained through ALE [149]. A challenge in ALE is to establish a genetic tool to quickly obtain the desired phenotype. To this end, smart microbial engineering could be implemented by designing suitable genetic circuits [150]. For example, by intergrading the pH sensor and riboswitch, a digital pH-sensing system RiDE was designed for the autonomous control of strain evolution. This universal platform can be used for rapidly obtaining succinic acid tolerance or lactic acid tolerance phenotypes [96].

### 4.3.2. Establishing Cell Mutagenesis

Compared to ALE, microbial mutagenesis technology enables strain mutagenesis in a faster and more efficient way to obtain the desired phenotype. Various strategies have been developed for strain mutagenesis, such as chemical or physical mutagenesis technology. For example, by using a combinational chemical (NTG) and plasma-based mutagenesis (ARTP) process, a *C. butyricum* mutant with tolerance to the high concentration of 1,3-PDO was obtained to produce 37 g·L$^{-1}$ 1,3-PDO, which was 29.48% higher than that of the wild strain [15]. The accumulation of organic acids would inhibit cellular growth and the target product production. To this end, an *L. reuteri* mutant with an increase of 18.6% in organic acid resistance was obtained through electron beam irradiation mutagenesis irrelevant. This mutant can produce 93.2 g·L$^{-1}$ 1,3-PDO, which was 34.6% higher than that of the wild-type strain [71].

### 4.3.3. Expressing Transporters

In addition, the expression of transporters is also an effective strategy to improve the tolerance of metabolite stress. By overexpressing transporters, the concentration of intracellular metabolites can be reduced to decrease the damage and death of cells [51]. For example, the toxic side effects of adipic acid on *S. cerevisiae* cells can be reduced by up-regulating the expression of specific multidrug resistance transporters Qdr3p (Figure 4c) [60].

### 4.3.4. In Situ Product Separation

The last method to improve microbial metabolite stress tolerance is in situ product separation. This method can rapidly reduce the concentration of metabolites in the bioreactor, thus improving the survival rate and production of microbial strains. Currently, there are two main approaches—membrane separation and medium exchange strategies—to achieve this goal. For example, by developing membrane separation-coupled fermentation technology, the feedback inhibition of propionic acid was reduced, which increased the concentration of succinic acid by 48.54% [97]. By establishing medium exchange strategies, the product inhibition of succinic acid was eliminated and the survival rate and succinic acid production of *Synechocystis* sp. PCC 6803 was improved [151].

In general, in the fermentation process, various internal or external factors, such as pH changes, temperature stress, osmotic press, oxidative stress, and metabolite accumulation, could impair efficient monomer bioproduction. To address this issue, different strategies, including adding protective agents, membrane engineering, expressing transporter or key genes, or ALE, were established to improve microbial environmental stress tolerance. Those rational or semi-rational strategies markedly shorten the time needed for increasing cell survival rate, metabolic activity, and microbial fitness, laying a solid foundation for efficient microbial biosynthesis.

## 5. Concluding Remarks

This review provides a systematic overview of how to improve bio-based monomer synthetic efficiency. Firstly, various metabolic engineering strategies were adopted to enable the efficient utilization of cheap substrates, so that microorganisms could efficiently utilize different substrates to decrease production cost; secondly, by regulating the enzyme, pathway, and metabolic network, the performance of the microbial cell factories can be improved to efficiently convert substrate into the target products; finally, different strategies such as protein engineering, gene expression regulation, and evolutionary engineering were implemented to enhance the stress tolerance of chassis cells when dealing with harsh environments or toxic metabolites.

While this review has illustrated the potential of metabolic engineering strategies and synthetic biology tools in boosting the production of bio-based monomers, there are some challenges that still need to be addressed: (i) Improving the yield of monomers. During the production of some monomers, such as dicarboxylic acids and diamines, the yield of products is low due to the decarboxylation reaction. To solve this problem, the waste carbon atom can be saved via two methods. On the one hand, the $CO_2$ fixation efficiency can be increased by mining novel carbon fixation enzymes [152], designing carbon fixation pathways with low energy demand [153], or building a light-powered system [154]. On the other hand, the $CO_2$ emission can be reduced by designing new synthetic pathways with decreased $CO_2$ release [155] or constructing a $CO_2$-mitigation molecular switch [62]; (ii) Mining the resources of non-model strains. Non-model strains that exhibited advantages over model strains can be explored for monomer production. For example, some dicarboxylic acid overproducers have evolved a high tolerance to high concentrations of dicarboxylic acids, such as *Aspergillus oryzae* for malic acid [156], *Pseudomonas* for cis,cis-muconic acid [157], and *A. succinogenes* and *M. succiniciproducens* for succinic acid production [158]. By establishing genetic manipulation tools for these non-model strains, efficient microbial cell factories can be built to address the current challenges in sustainability.

**Author Contributions:** Conceptualization, C.G., C.C., X.C., J.W. and L.L.; Writing—original draft preparation, C.C.; Writing—review and editing, C.G. All authors have read and agreed to the published version of the manuscript.

**Funding:** This work was supported by the National Key R&D Program of China (2021YFC2100700), the Science Fund for Creative Research Groups of the National Natural Science Foundation of China (32021005), and the Tianjin Synthetic Biotechnology Innovation Capacity Improvement Project (TSBICIP-KJGG-015).

**Institutional Review Board Statement:** Not applicable.

**Informed Consent Statement:** Not applicable.

**Data Availability Statement:** Data sharing not applicable.

**Conflicts of Interest:** The authors declare no conflict of interest. The funders had no role in the design of the study; in the collection, analyses, or interpretation of data; in the writing of the manuscript; or in the decision to publish the results.

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
