# Peer review of "Engineering Microorganisms to Produce Bio-Based Monomers: Progress and Challenges"

_fermentation, doi:10.3390/fermentation9020137_

Round 1
Reviewer 1 Report
Peer review for Chen et al. – “Engineering microorganisms to produce bio-based monomers: progress and challenges”
Summary:
This article reviews approaches for microbial production of monomers that can be incorporated into bioplastics and other polymers, with the aim of reducing dependence on petroleum-derived products. Engineering targets – namely, substrate utilization, pathway development, and stress tolerance – are identified and discussed with recent examples, and some challenges and opportunities for the future are presented. This review provides a relevant contribution to the field, but would benefit from the following minor modifications before it is fit for publication in Fermentation:
Comments:
1. Lines 33-64: Instead of presenting the production data (market share, product volume, etc.) in text form, this information could be presented as a table. I think this would make it more accessible for readers.
2. In several instances, the authors re-state the thesis of the review at the beginning of a paragraph. I find this unnecessary and redundant, and these phrases should be removed for concision:
a. Lines 65-68: “Currently, plastics are…sustainable development.”
b. Lines 92-94: “To build…sound bioprocesses.”
c. Lines 111-114: “It is of great significance…bio-based monomers.”
d. Lines 609-611: “With the development…bringing positive effects;”
3. Lines 76-77: “The raw materials of bio-based plastics mostly come from food crops…” this may have been the case 10-20 years ago, but is it still true? I would revise this statement to emphasize the need for renewable bioplastics fermentation substrates, which includes avoidance of food crops and/or restructuring of infrastructure to acquire renewable raw materials (which is not currently as efficient as that for petroleum).
4. Several claims require justification with more references, including:
a. Lines 49-51: “Succinic acid has a wide range of applications…” should include a reference that reviews the industrial applications for this compound.
b. Line 146-147: “The recycling and reuse of food waste…circular economy.” Could use a reference to a review on this topic, e.g. Lad et al., J Ind Microbiol Biotechnol (2022).
c. Lines 184-186: “The introduction of…efficiency of material monomers.” Should provide example(s) of heterologous expression inducing a metabolic burden on cells.
d. Lines 253-255: The statement is made that lignocellulose pretreatment releases growth inhibitors, but no references are provided.
e. Lines 433-434: “To do so,… carry out directed evolution.” There are many reviews on the use of directed evolution for production of biochemicals that should be cited here, e.g. Porter et al., ChemBioChem (2016).
5. Line 123: “yeast” should not be italicized.
6. Line 203: “Detoxicating” should be changed to “Detoxifying”
7. Lines 206-207: Should be modified to emphasize that some organisms cannot utilize polymeric sugars/substrates, since this is what is discussed in the paragraph.
8. Lines 263-276: this section on enhancing strain growth performance should include some discussion of media composition and reactor design as important influencers of bioproduction rate.
9. Lines 286-294: it should be noted that transporter overexpression can be a risky strategy to increase uptake; too many copies of transporter proteins may lead to membrane disruption and poor cell viability. Strategies should therefore be considered to tune transporter expression levels appropriately.
10. Lines 295-308: another important method to alleviate carbon catabolite repression is to identify and delete carbon repression control (crc) genes, as demonstrated in Johnson et al., Metab Eng Commun (2017).
11. Figure 2a: the disulfide bond drawing should include sulfhydryl groups, not amine groups.
12. Lines 395-407: it should be mentioned that expression of heterologous enzymes can also help alleviate bottlenecks introduced by cofactor limitations of endogenous enzymes. E.g., using a cofactorless enzyme from a heterologous source could liberate the native cofactor for participation in other important reactions, increasing strain efficiency.
13. Lines 422-429: it should be noted that tunability of protein expression levels requires functional genomic modification tools in the desired microbial cell factory; some species are more suited for genomic modification than others.
14. The authors mention metabolic fluxes, enzyme expression levels, and regulatory elements as targets to improve bioproduction efficiency, but little is said about how these targets are identified (e.g., adaptive laboratory evolution, omics studies, etc.).
15. Line 630: “The pH values of the fermentation system will change along with the accumulation of products.” Possible reasons for this phenomenon (e.g. production of an acidic product such as succinate) should be explained.
16. Lines 722-729: besides using amino acids as a media supplement to improve osmoprotection, strains could be engineered to accumulate these compounds intracellularly.
Reviewer 2 Report
This review summarizes the progress and challenges in microbial production of bio-based monomers which could be an important reference for the researchers working on the same field. However, there are issues that need to be addressed before the publication of this paper.
Comments:
1. The introduction section was very confusing and quite misleading, when I first read this manuscript. Since this section done a great job of describing the basic importance of bioplastic production and the economic importance of bio-based monomers. The theme of this introduction had little relationship to the title and the main text. I suggest re-writing the introduction to pull together the body of the paper (the title) and the introduction to give emphasis to the progress and challenges in the development of cell factories for bio-based monomers.
2. Please avoid using the term “more and more”. Please change it to simply “more”.
3. Some description in the introduction is redundant. For example, lines 65-68 describing the importance of bioplastic was already described in the first paragraph of the introduction. Please avoid repeating what has been described/discussed already. This is similar for lines 92-94.
4. Please add citations for lines 73-74, 170 ( describing parallel metabolic pathway), 184-186, 198-200, and 206-207.
5. It is also important to cite the figures throughout the manuscript.
6. Add figures that describe substrates depolymerization, relieving carbon catabolite repression, co-utilizing multi-substrate, dynamic pathway regulation through biosensors, regulating cellular metabolic networks, decoupling cell growth and production, engineering transcription factor, cell membrane engineering, adaptive laboratory evolution, and tuning redox balance.
7. The review is lacking accurate data (the titer or the improved fold) when described mostly previous studies throughout the manuscript. Alternatively, the review lacks critically evaluate of the available scientific data from previous work. I suggest adding a table that shows the titer, yield, and productivity of bio-based monomers as well as the metabolic engineering or synthetic biology techniques to produce them.
8. It is also important to show the metabolic pathways (as one additional figure) of the major biobased-monomers from substrates that are described in this review manuscript.
9. Elements in figure 1b is crowded, please redraw to make it clearer to the reader. The C5 and C6 sugars should be indicated. It would also be better if the backgrounds for all the figure is white.
Round 2
Reviewer 2 Report
My points were properly addressed. Accept as updated.
Author Response
Thank you very much.